# A Study on the Similarities and Differences of the Conventional Gasoline Spot Price Fluctuation Network between Different Harbors



**Guangyong Zhang [1], Lixin Tian [1,2,*], Wenbin Zhang [1,3], Xu Yan [1], Bingyue Wan [1] and Zaili Zhen [1]**

1   Energy Development and Environmental Protection Strategy Research Center, Jiangsu University, Zhenjiang 212013, China; zhanggy_ujs@126.com (G.Z.); zubnust@163.com (W.Z.); yanxu_16@163.com (X.Y.); wby618906@126.com (B.W.); lddcb@126.com (Z.Z.)
2   Energy Interdependence Behavior and Strategy Research Center, School of Mathematical Sciences, Nanjing Normal University, Nanjing 210046, China
3   School of Mathematical Science, Taizhou Institute of Science and Technology, Taizhou 225300, China
*   Correspondence: tianlx@ujs.edu.cn; Tel.: +86-135-0528-4660

**Abstract:** According to the fluctuation series of the conventional gasoline spot prices (CGSP) in New York Harbor (NYH) and U.S. Gulf Coast (GC), this paper defines the fluctuation modes by the coarse-grained method based on the CGSP series in the two harbors. The fluctuation series are converted into the characters by means of the sliding window, where five symbol series is used as a fluctuation mode, one day was used as a step to slide in the data window, and the conventional gasoline spot prices fluctuation network (CGSPFN) is constructed in the two harbors. Then the evolutionary rule of the new nodes in the CGSPFN is analyzed, such as the strength and distribution, average shortest paths, conversion cycle, betweenness, and clustering coefficient of the nodes are calculated in different periods. The result indicates that the cumulative time of the new nodes which appeared in the CGSPFN is not random but presents a high linear growth trend, which reveals the linear features of the cumulative time of abnormal points when the gasoline price fluctuation appears. The betweenness and clustering coefficient shows that the nodes with the larger strength have smaller betweenness and clustering coefficients, the nodes with the larger betweenness have smaller strength and clustering coefficients, and the nodes with the larger clustering coefficients have smaller betweenness and strength. Meanwhile, the gasoline prices are in a transitional period when the larger indicators appear and have a rising trend, and identifying the transitional period will help the decision maker to grasp the regularity of the changes of the gasoline prices.

**Keywords:** gasoline prices; topological properties; network structure; fluctuation mode

## 1. Introduction

Gasoline is one of the most consumed petroleum products and an important fuel for engines, which is closely related to the transportation industry [1,2]. In the United States, known as "Automobile Kingdom", the fluctuations of the gasoline prices not only related to social welfare [3] and subjective well-being [4], but also had a strong correlation with macroeconomic fluctuations [5]. In addition, the gasoline prices had a great impact on the urban and rural traffic safety [6] and traffic behavior [7]. The final consumption prices of the gasoline consist of four parts: crude oil costs, refining costs and profits, sales and transportation costs, and taxes, in which crude oil costs, oil refining costs and profits had a significant impact on the fluctuation of the gasoline price [8]. At the same time, the external conditions of the mutation (such as war, bad weather, etc.) will also lead to fluctuations of the gasoline prices [9,10]. Rising fuel prices have dampened private drivers' demand for gasoline in the U.S.,

making the U.S. market dependent on sustained economic and freight expansion to stimulate oil use. According to the Federal Highway Administration in 2018, U.S. traffic volumes were up by just 0.3% on a seasonally adjusted basis in the three months from April to June compared with the same period a year earlier. According to the U.S. Energy Information Administration, slower traffic growth has been mirrored in flattening gasoline consumption, retail gasoline prices are up by more than 55% from their cyclical low in February 2016, and the average daily demand for automotive gasoline is 9.201 million barrels in March 2019, which is 1.5% lower than the same period last year. These factors have an impact on the gasoline prices. Therefore, how to reveal the fluctuation mechanism of the gasoline prices has attracted much attention.

The asymmetry of the gasoline price fluctuation has been around for a long time. In 1991, Bacon [11] researched the "rockets and feathers" hypothesis about the different change of the gasoline prices to increased and decreased cost, which showed that the change of the retail gasoline prices was slightly faster when the retail gasoline cost increased. Then, Galeotti [12] re-examined "rocket and feathers" for the international comparison of the European gasoline markets, i.e., the asymmetric transmission about the fluctuation of the crude oil price to the retail gasoline prices, which indicated that the increase of the gasoline prices was faster when the cost of the retail gasoline prices increased in the United States [13]. Douglas [14] detected the sensitivity of the results to abnormal data by estimating a threshold cointegration model with multiple mechanisms. Meanwhile, Bremmer [15] discussed the relationship between the prices of the retail gasoline and crude oil in the United States during the Great Recession, and proposed the "balloons and rocks" behavior which was contrary to the "rocket and feathers" behavior. In addition, the correlation analysis of the retail gasoline prices in different regions was carried out to test the hypothesis that there was no strong cross-correlation or anti-correlation by using multi-scale cross-correlation analysis method [16]. These researches showed that there is a symmetric about the fluctuation of the gasoline prices.

Data released by the U.S. Energy Information Agency in January 2020 shows that U.S. crude oil production has more than doubled in the past 10 years to 12.66 million barrels per day. The domestic gasoline prices are increasing continuously with the soaring prices of the international crude oil in recent years, which has caused a significant impact, and all sectors are highly concerned about the fluctuation of crude oil prices on gasoline prices in the United States. According to the study on the crude oil prices to the gasoline prices, it is found that the gasoline prices asymmetrically responded to the increase and decrease of the crude oil prices, and there was a positive impact with the crude oil prices to the gasoline prices [17]. Other researches also showed that there were some differences with the market of the gasoline prices in different areas [18], and the asymmetric transmission with the crude oil prices to the gasoline prices mainly occurred in the downstream of the transmission process rather than the upstream [19]. The multi-factor study on the gasoline prices showed that the implementation of biofuel policy, transportation demand and income, adjustment of wholesale gasoline market structure and gasoline reserves, integration of gasoline industry, the changes of the regional tax and other behaviors had different impacts on the gasoline price fluctuations [20–28]. In addition, gasoline reserves in the United States have reached 259 million barrels with the largest surplus in 27 years, and the demand for gasoline has decreased. Therefore, there will be downside risks to gasoline prices if gasoline demand does not rebound; reason being is that if gasoline demand no longer rebounds, gasoline reserves will rise further. By then, refineries will have to reduce output and their crude oil purchases will decline. Declining crude oil sales will lead to rising crude oil reserves, which will reduce crude oil prices. Eventually, the supply will decrease. In general, if the gasoline demand in the United States and even the world cannot be recovered in the short term, there will be downside risks to oil prices this year.

As for the time series, there are many research methods. Except for the traditional econometric methods for the time series, such as error correction models [29], autoregressive moving average with exogenous variable model with autoregressive conditional heteroskedasticity [30], and autoregressive conditional heteroskedastic models [31] are used. Additionally, the complex network methods have

been used to research on the field of the energy and economics by means of the time series data, and have received extensive attention with the rapid development of the complex network. An [32] designed a method to examine the dynamics of the co-movement between futures and spot prices of the crude oil. Wang [33] presented the phase space coarse- graining algorithm, which converts a time series into a directed and weighted complex network, and researches the fluctuation behavior of the crude oil and gasoline price based on these methods [34]. According to the methods, Chen [35] analyzed the dynamic evolutionary behavior of the spot and futures price of the heating oil. In the real economic activity, a large number of complex systems can be analyzed and used to construct a network by processing the data, and there are also many valuable researches on the problem of the oil prices by using the complex network methods [32,36–40]. All these studies have obtained good results, which can clearly be seen that the complex network method is not only a manifestation of the data, but also an important tool of the scientific research for the time series data. Therefore, this paper is based on the research method of complex network theory, and the way of data processing and the spatial-temporal distribution characteristics of research objects are different from previous studies.

The above studies are based on the fluctuation rules between gasoline prices and other product prices, but less researches on spatial fluctuation rules, the difference about the sensitivity of gasoline prices and relevance of gasoline prices in different harbors. Based on the fluctuation information of the CGSP between different harbors, this paper defines the network nodes after a coarse-grained algorithm is used to the data, and then the directed and weighted network models of the CGSP are established in different harbors, then the network topological properties that involve the spectra of networks, node strength and its distribution, the average path length etc., are compared and analyzed in different periods. This study aims to explore the following issues:

What are the rules and mechanisms of gasoline price fluctuation in different harbors?

Is there a difference about the sensitivity of gasoline prices under the same external stimulation in different harbors, and how lagging is the gasoline price fluctuation?

From the perspective of the time when the new node appears, is the change in the gasoline price fluctuation series regular?

What are the similarities and differences of the gasoline price fluctuation in different harbors, and what is the relationship between them?

The structure of this paper is organized as follows: Section 2 introduces the data and method, which involves the source and processing of the data, the period division and the construction process of the conventional gasoline spot prices fluctuation network (CGSPFN). In Section 3, we analyze the topological properties of the CGSPFNs of different periods in New York Harbor (NYH) and U.S. Gulf Coast (GC). A comprehensive discussion is proposed in Section 4. In Section 5, the summaries and implications are given.

## 2. Data and Methods

### 2.1. The Source and Processing of the Data

In this paper, the data is derived from the New York Harbor Conventional Gasoline Regular Spot FOB Price (dollars per gallon) and U.S. Gulf Coast Conventional Gasoline Regular Spot FOB Price (dollars per gallon) from July 14, 1986 to May 31, 2018 by the U.S. Energy Information Administration. In order to improve the effectiveness and completeness of the data, we utilize the mean of the previous and next data as a supplement for the missing parts of the data, and the trends of their change and the relative D-values are shown in Figure 1.

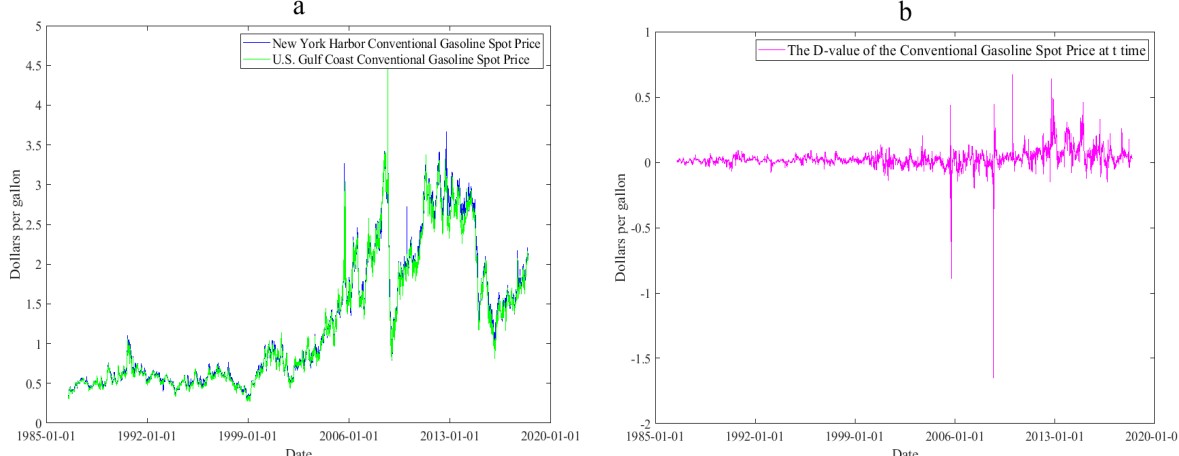

**Figure 1.** (**a**) the fluctuation trend of the conventional gasoline spot prices (CGSP) in New York Harbor (NYH) and U.S. Gulf Coast (GC); (**b**) the trend of change in the relative D-values of the CGSP between NYH and GC.

The CGSP series in NYH and GC are recorded as $P_{NYH}(t)$ and $P_{GC}(t)$, $(t = 1, 2, 3, \cdots, 8319)$, respectively. As for the CGSP in NYH, the focus is the information of the changing price in this paper, which is assumed that $P_{NYH}(t)$ is the CGSP in the t-period, then the fluctuation series of the CGSP are recorded as $\Delta P_{NYH}(t) = P_{NYH}(t) - P_{NYH}(t-1)$. Let $M_{\Delta P_{NYH}} = \frac{4 \sum_{t=1}^{n-1} |\Delta P_{NYH}(t)|}{5(n-1)}$, then $\Delta P_{NYH}(t) > M_{\Delta P_{NYH}}$ represent the fast increase of the gasoline prices, $0 < \Delta P_{NYH}(t) \leq M_{\Delta P_{NYH}}$ represent the slow increase of the gasoline prices, $M_{\Delta P_{NYH}} = 0$ represent the steady fluctuation of the gasoline prices, $-M_{\Delta P_{NYH}} \leq \Delta P_{NYH}(t) < 0$ represent the slow decrease of the gasoline prices, and $\Delta P_{NYH}(t) < -M_{\Delta P_{NYH}}$ represent the fast decrease of the gasoline prices.

In order to reveal the rules of the CGSP between NYH and GC more clearly, and let each $\Delta P_{NYH}(t)$ correspond to a symbol of $SP_j$, we obtain the CGSP series, as shown in Formula (1).

$$SP_j = \begin{cases} I, \Delta P_{NYH}(t) > M_{\Delta P_{NYH}} \\ i, 0 < \Delta P_{NYH}(t) \leq M_{\Delta P_{NYH}} \\ e, M_{\Delta P_{NYH}} = 0 \\ d, -M_{\Delta P_{NYH}} \leq \Delta P_{MYH}(t) < 0 \\ D, \Delta P_{NYH}(t) < -M_{\Delta P_{NYH}} \end{cases} \tag{1}$$

where $I, i, e, d$ and $D$ indicate the fast increased fluctuation, the slow increased fluctuation, the steady fluctuation, the slow decreased fluctuation and the fast decreased fluctuation, respectively. Thus, the CGSP fluctuation series are transformed into the corresponding symbol series in NYH, as shown in Formula (2).

$$PP_{NYH} = \{HP_1, HP_2, HP_3, \cdots\}, HP_j \in (I, i, e, d, D). \tag{2}$$

Similarly, the CGSP fluctuation series are transformed into a corresponding symbol series in GC, as shown in Formula (3).

$$PP_{GC} = \{HP_1, HP_2, HP_3, \cdots\}, HP_j \in (I, i, e, d, D). \tag{3}$$

## 2.2. Dividing the Period

One year is used as a period, the probability of the symbols $I, i, e, d, D$ is counted in each period, and recorded as $PI, Pi, Pe, Pd, PD$, respectively. Thus, we obtained the evolutionary image of the probability for the CGSP fluctuation state in NYH, as shown in Figure 2.

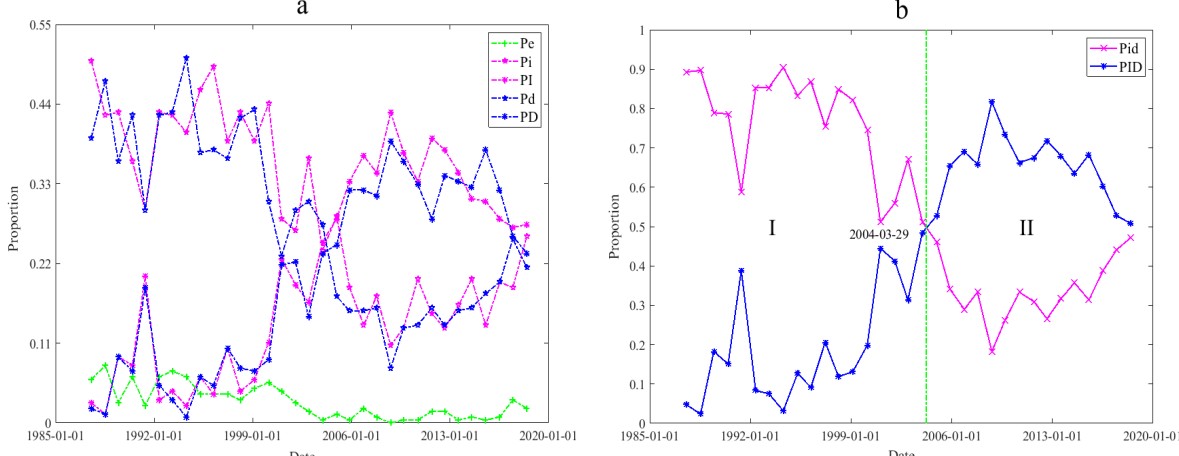

**Figure 2.** (**a**) the probability evolutionary image of the *Pe, Pi, PI, Pd, and PD* of the CGSP in NYH; (**b**) the evolutionary image of the probability for the *Pid* and *PID* in NHY.

According to Figure 2a, the probability of the fast increased state (*PI*), the slow increased state (*Pi*), the stable state (*Pe*), the slow decreased state (*Pd*) and the fast decreased state (*PD*) alternately appear in 2004. In order to further explore the fast fluctuation and the slow fluctuation of the CGSP in NYH, the following calculations are given:

$$Pid = Pi + Pd, PID = PI + PD \tag{4}$$

where the *Pid* indicates the probability of the slow fluctuation state and *PID* indicates the probability of the slow fluctuation state of the CGSP in NYH. Thus, we obtain the evolutionary image of the probability of the CGSP fluctuation state *Pid* and *PID* in NYH, as shown in Figure 2b. According to Figure 2b, we can find that the probability of the fast fluctuation state is greater than the probability of the slow fluctuation state from 29 March, 2004, which shows that the fluctuation states of the CGSP have changed at that time. Therefore, we divided the period into the slow fluctuation period (SFP) and fast fluctuation period (FFP), as shown in Table 1.

**Table 1.** The period division of the CGSP fluctuation in NYH.

| Serial Number | Time Interval | Fluctuation Periods |
| --- | --- | --- |
| I | 14 July, 1986~29 March, 2004 | SFP |
| II | 30 March, 2004~31 May, 2018 | FFP |

By means of the above method dividing the period, the evolutionary image of the probability of the CGSP fluctuation state can be obtained in GC, as shown in Figure 3.

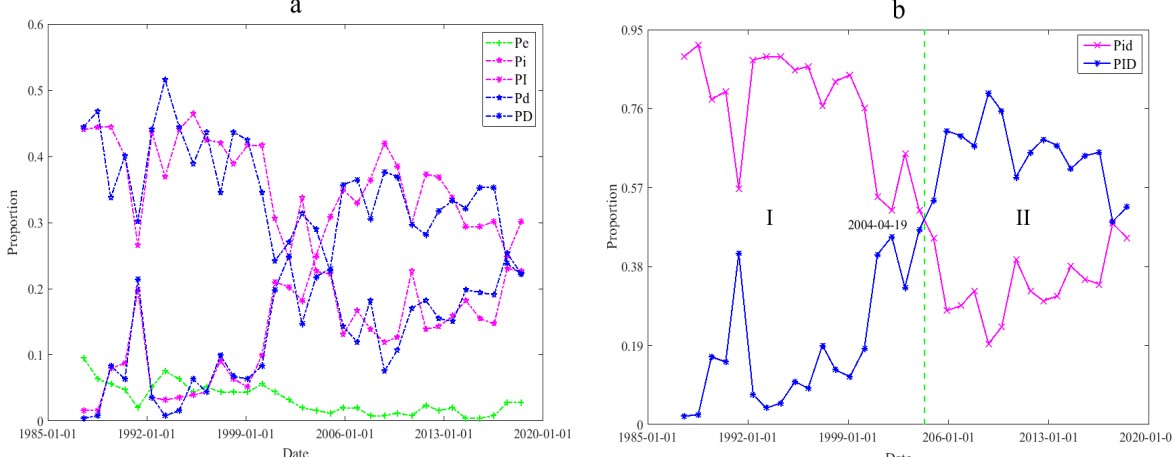

**Figure 3.** (**a**) the probability evolutionary image of the *Pe, Pi, PI, Pd, and PD* of the CGSP in GC; (**b**) the evolutionary image of the probability for the *Pid and PID* in GC.

On the basis of Figure 3, the SFP and FFP is also divided in GC, as shown in Table 2.

**Table 2.** The period division of the CGSP fluctuation in GC.

| Serial Number | Time Interval | Fluctuation State |
|---|---|---|
| I | 14 July, 1986~19 April, 2004 | SFP |
| II | 20 April, 2004~31 May, 2018 | FFP |

Certainly, the increase period and decrease period can be divided if the fluctuation states of the SFP and FFP are made in a deep research in NYH and GC. That is, we have the following Formula (5):

$$PiI = Pi + PI, PdD = Pd + PD \tag{5}$$

where *PiI* indicates the probability of the increase state of the CGSP and *PdD* indicates the probability of the decrease state of the CGSP. Then we can get more detailed periods, such as the fast increased period, slow increased period, fast decreased period and slow decreased period. However, we mainly researched the network properties of the whole period, SFP and FFP, but the more detailed periods about Formula (5) are not presented in this paper.

*2.3. The Constructing Networks*

2.3.1. The Constructing Process of the Networks

The resolution of the time series is also different when the time interval is chosen differently in the process of the CGSP series transformed into the symbol series. Thus, the length of the symbol series, the occupancy of each symbol, and the correlation among the symbols will be different for the same time series. In order to keep the consistency between the theoretical analysis and the actual time interval, five symbol series are used as a fluctuation mode and one day is used as a step to slide in the data window. In addition, there is transitivity and tropism among the fluctuation modes and the reason is that the fluctuation modes are formed by the sliding data, i.e., the next fluctuation mode is formed on the basis of the previous fluctuation states. Therefore, a directed and weighted network of the CGSP fluctuation can be constructed in NYH and GC, where the fluctuation mode, the direction and the times of transformation are defined as the node, the edge, and the weight of the node in the CGSPFN. The CGSP in NYH is used as an example, the constructing process of which is shown in Table 3.

**Table 3.** The constructing process of the directed and weighted network of the CGSP fluctuation.

| Date | $P_{NYH}(t)$ | $\Delta P_{NYH}(t)$ | $PP_{NYH}(t)$ | Fluctuation Mode |
|---|---|---|---|---|
| 1986-07-14 | 2.053 | | | |
| 1986-07-15 | 2.031 | 0.019 | *I* | |
| 1986-07-16 | 2.080 | 0.006 | *i* | |
| 1986-07-17 | 2.142 | −0.009 | *d* | |
| 1986-07-18 | 2.133 | 0.009 | *i* | |
| 1986-07-21 | 2.163 | 0.005 | *i* | *Iidii* |
| 1986-07-22 | 2.167 | −0.021 | *D* | *idiiD* |
| 1986-07-23 | 2.174 | 0.007 | *i* | *diiDi* |
| 1986-07-24 | 2.162 | −0.005 | *d* | *iiDid* |
| 1986-07-25 | 2.177 | −0.009 | *d* | *iDidd* |
| 1986-07-28 | 2.172 | −0.013 | *d* | *Diddd* |
| 1986-07-29 | 2.125 | 0.007 | *i* | *idddi* |
| 1986-07-30 | 2.152 | 0.014 | *i* | *dddii* |
| 1986-07-31 | 2.146 | −0.020 | *D* | *ddiiD* |
| 1986-07-14 | 2.138 | 0.009 | *i* | *diiDi* |
| … | … | … | … | … |
| 2018-05-31 | 2.093 | −0.029 | *D* | *DeDID* |

Sliding Window

According to the method in Table 3, we can obtain 8314 fluctuation modes of the CGSP in NYH and GC, which involve the same fluctuation modes, i.e., the nodes are {*Iidii*, *idiiD*, ⋯ , *DeDID*}. Therefore, the node of the network can be obtained after the same nodes are deleted, then the CGSPFNs are constructed.

### 2.3.2. The CGSPFN in NYH and GC

In order to present the complete features of the network, we utilized the full data to construct the CGSPFN of different periods in NYH and GC. Then, the CGSPFNs are obtained in different periods in NYH and GC, as shown in Figure 4. Furthermore, the different periods involve the whole period, SFP and FFP.

(a)　　　　　　　　　　　　　　　　　　　　(b)

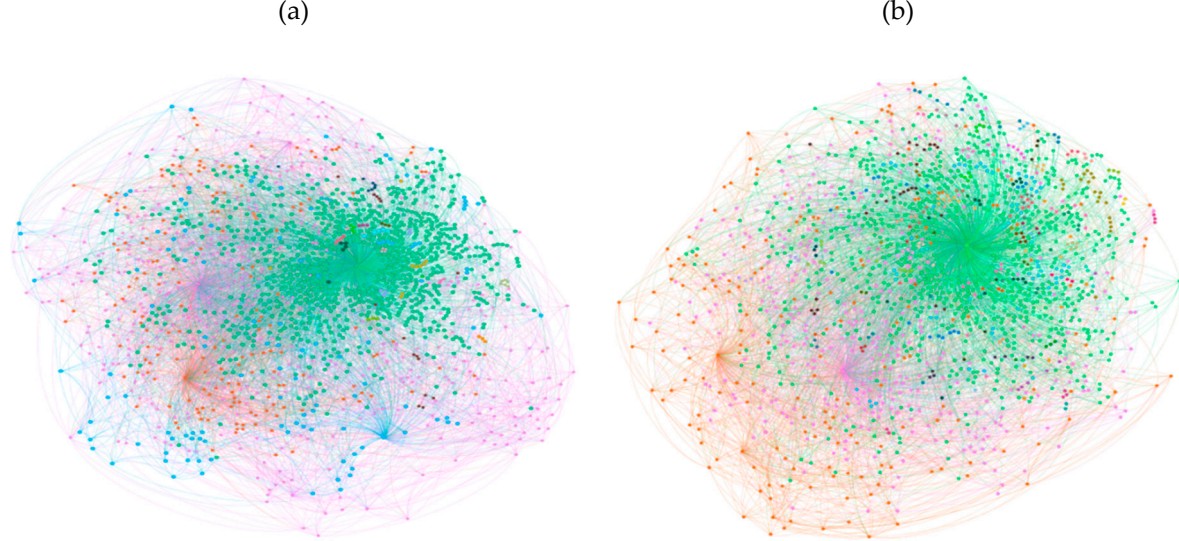

**Figure 4.** *Cont.*

(c)　　　　　　　　　　　　　　　　　　　　　(d)

(e)　　　　　　　　　　　　　　　　　　　　　(d)

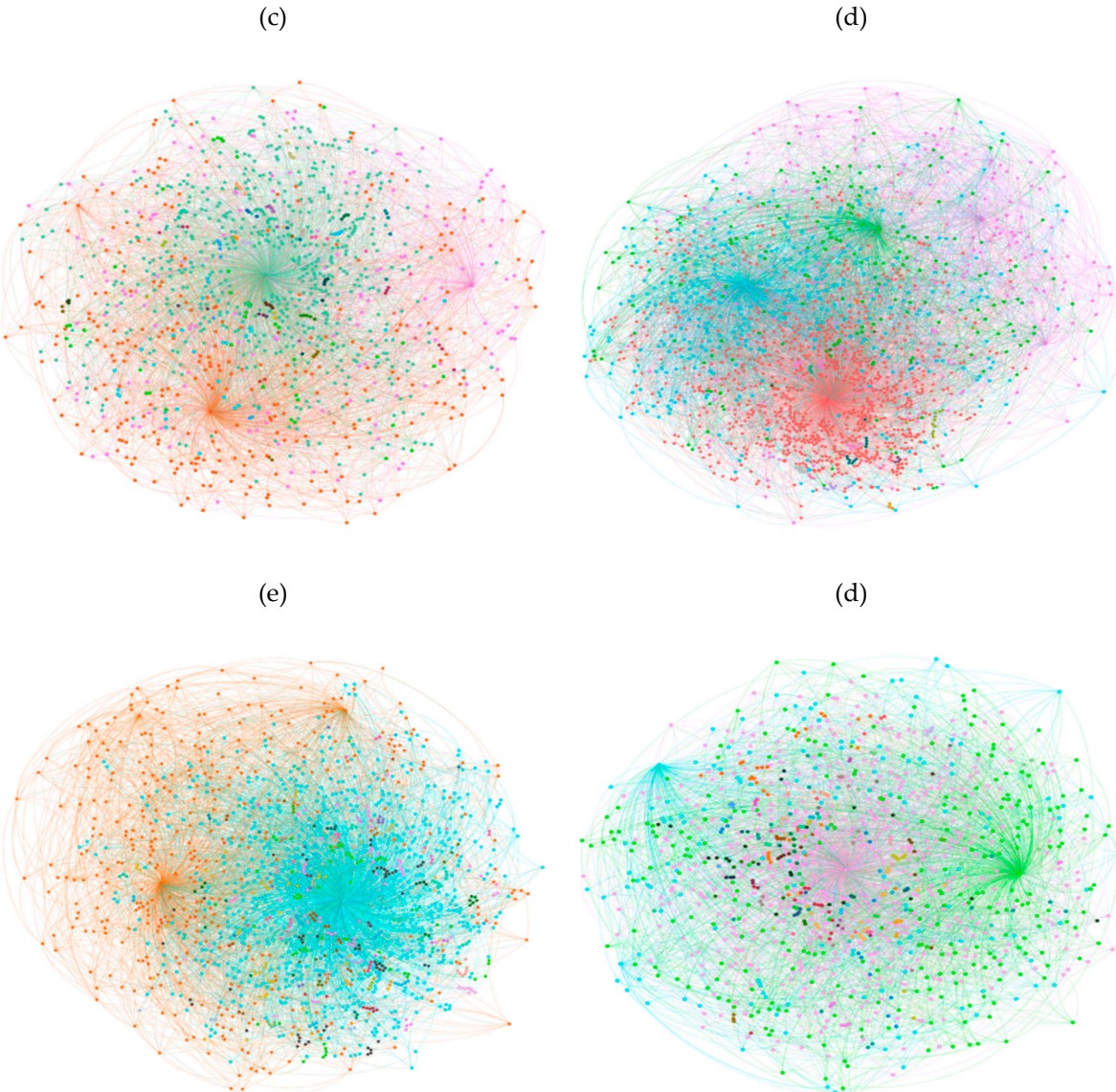

**Figure 4.** The CGSPFN of different periods in NYH and GC: (**a**) the CGSPFN of the whole period in NYH; (**b**) the CGSPFN of the SFP in NYH; (**c**) the CGSPFN of the FFP in NYH; (**d**) the CGSPFN of the whole period in GC; (**e**) the CGSPFN of the SFP in GC; (**f**) the CGSPFN of the FFP in GC.

As shown in Figure 4a,c, we can see that there is a higher complex among the connections of different nodes of the CGSPFN in the whole period in NYH and GC. According to Figure 4b,c,e,f, the CGSPFNs have different network structures in the SFP and FFP, which means that some nodes show strong intermediary, but most and sub-nodes show weak connectivity.

According to the CGSPFN in different periods in NYH and GC (Figure 4), the fluctuation modes are consisted with five symbols. Therefore, there are 3125 ($5^5$) different node types in the theory. Based on the above analysis, we obtain the number of the different node types, as shown in Table 4.

**Table 4.** The number of the different node types in different periods in NYH and GC.

| Different Periods / Different Harbors | NYH | GC |
|---|---|---|
| The whole period | 1592 | 1634 |
| The SFP | 1631 | 1646 |
| The FFP | 1157 | 1210 |

As for the number of the different node types in different periods in NYH and GC, the number in the SFP is more than that in the FFP, and the reason is that the Formula (6) is different in different periods, which is determined by the actual data of each period.

$$M_{\Delta P_{harbor}} = \frac{4 \sum\limits_{t=1}^{n-1} \left| P_{harbor}(t) - P_{harbor}(t-1) \right|}{5(n-1)} \tag{6}$$

where $P_{harbor}(t)$ is the current price of the harbor and $P_{harbor}(t-1)$ is the previous price in the harbor. The situation will not appear if we utilize the unified $M_{\Delta P_{harbor}}$ in each period. In addition, the number of the different node types of the CGSPFN in GC is more than that in NYH in a corresponding period. According to the above consideration, we can infer that the fluctuation state of the CGSPFN is more complex.

## 3. Analysis of the CGSPFN in NYH and GC

### 3.1. Analysis of the Correlation for the CGSPFN

Many scholars have conducted a systematic study and believe that there is a close correlation about the gasoline prices in different areas [16,18]. In this section, we will explore the correlation based on the nodes of the two networks. According to the above results, the number of the nodes of the CGSPFN is 1592 and 1634 in NYH and GC, respectively. The number of the same nodes is 1288 among the network nodes. The same node strength is counted in the whole period. According to the above results, the following Formula (7) is applied to calculate the similarity of the CGSPFN in NYH and GC.

$$l = \frac{\sum\limits_{i=1}^{n} \left( s_i^{NYH} - \bar{s}^{NYH} \right) \left( s_i^{GC} - \bar{s}^{GC} \right)}{\sqrt{\sum\limits_{i=1}^{n} \left( s_i^{NYH} - \bar{s}^{NYH} \right)^2 \sum\limits_{i=1}^{n} \left( s_i^{GC} - \bar{s}^{GC} \right)^2}} \frac{2M_{same}}{M_{NYH} + M_{GC}} \tag{7}$$

where $M_{NYH}$ and $M_{GC}$ denote the number of the node of the CGSPFN in NYH and GC, respectively. $M_{same}$ denotes the number of the same nodes of the CGSPFN in NYH and GC. Meanwhile, $0 \le l \le 1$, when $l = 0$, is shown that the similarity is the lowest between the two networks; when $l = 1$, it is shown that the similarity is the highest between the two networks. According to the calculation, we can obtain that the correlation of the node strength is 0.7405 between the two CGSPFNs, which shows there are close relationships of the strength of the same nodes of the CGSPFN and there is a higher similarity in NYH and GC.

In terms of the SFP and FFP, the correlation is also explored about the nodes of the CGSPFN based on Table 4. The number of the same nodes is 1222 in the SFP, but it is 966 in the FFP. According to Formula (7), the network similarity in the SFP is 0.4986, but only 0.3069 in the FFP, which illustrates that the CGSPFNs have a higher interdependence between the NYH and GC in the SSP than in the FFP.

By means of the comprehensive analysis, the interdependence between the CGSP fluctuations of the NYH and GC can be described by means of the network similarity, which shows that there is

a higher interdependence between the two networks in the whole period, but the interdependence gradually decreases in SFP and FFP.

### 3.2. Analysis of the Regularity for the New Nodes

Based on the CGSPFN of different harbors in different periods (i.e., the whole period, the SFP and the FFP), we explore the regularity of the cumulative time interval for the new nodes of the network. As for the new nodes, the same number of the nodes is chosen from the first appeared node, which means the same number of the nodes is decided by the least number of the nodes in the network. Then, we obtain the regularity of the cumulative time interval for the new nodes in NYH and GC, as shown in Figure 5.

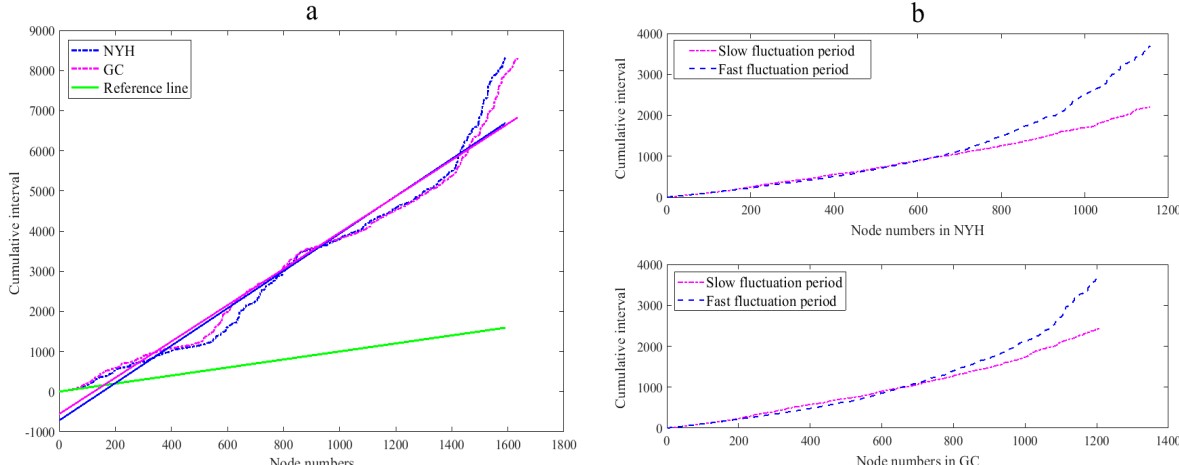

**Figure 5.** (**a**) the cumulative time interval of the new nodes in the whole periods in NYH and GC; (**b**) the cumulative time interval of the new nodes in the SFP and FFP in NYH and GC.

In Figure 5a, the green line represents the equal interval curve, and the new nodes of the CGSPFNs appear at unequal intervals as the time goes on, but gradually increase, and it shows a trend of the linear growth. The least square method is utilized to regress the cumulative time interval for the new nodes of the CGSPFN in NYH and GC, and the corresponding regression equations are $y = 4.6570x - 719.8704$ and $y = 4.5201x - 559.6975$, respectively. Meanwhile, the corresponding coefficients of determination ($R^2$) of the trend line are 0.9611 and 0.9674. Obviously, the result has a higher credibility and the cumulative time interval for the new nodes of the CGSPFN has a linear growth trend and shows a good regularity in NYH and GC, which reflects that the two networks have the similarity and the foresight of the CGSP under the temporal distribution.

In Figure 5b, the image of the cumulative time interval for the new nodes is presented. It can be seen that the cumulative time intervals for the new nodes of the CGSPFN also have a good regularity in the SFP and FFP, i.e., a trend of the linear growth. The corresponding regression equations and coefficients of determination are shown in Table 5.

**Table 5.** The regression equation, coefficient of determination, and P-Value of the cumulative time interval for the new nodes of the CGSPFN in the SFP and FFP.

| Different Harbors | Periods | Regression Equation | $R^2$ | P-Value |
|---|---|---|---|---|
| NYH | The SFP | $y = 1.8376x - 145.9131$ | 0.9851 | 0.0000 |
| | The FFP | $y = 2.8120x - 475.1978$ | 0.9001 | 0.0000 |
| GC | SFP | $y = 1.9417x - 179.0398$ | 0.9789 | 0.0000 |
| | FFP | $y = 2.6350x - 450.9030$ | 0.9050 | 0.0000 |

According to Table 5, the result has a higher credibility, which illustrates that the cumulative time interval for the new nodes of the CGSPFN has a trend of the linear growth, very high similarity and a good regularity in the SFP and FFP. In addition, the cumulative time interval for the new nodes in the FFP is larger than that in the SFP, as shown in Figure 5b.

To sum up, the new nodes of the CGSPFN mean that the price of the abnormal fluctuations has appeared and the new nodes are different from the previous fluctuations; and the CGSP has complex nonlinear features in NYH and GC, but the cumulative time interval for the price of abnormal fluctuations shows a trend of the linear growth. The time can be identified effectively by the regularity when the price of the abnormal fluctuation appears, which can help the decision makers to respond to the fluctuations of the new price nodes in time, so as to improve the accuracy of the judgment and reduce the risk for the CGSP in NYH and GC.

### 3.3. Analysis of the Node Strength and its Distribution

In order to further reveal the temporal distribution characteristics and power law distribution of the important nodes for the CGSPFN in NYH and GC, the strength [41] and its distribution [32] about the nodes are explored in the whole period, as shown in Figure 6.

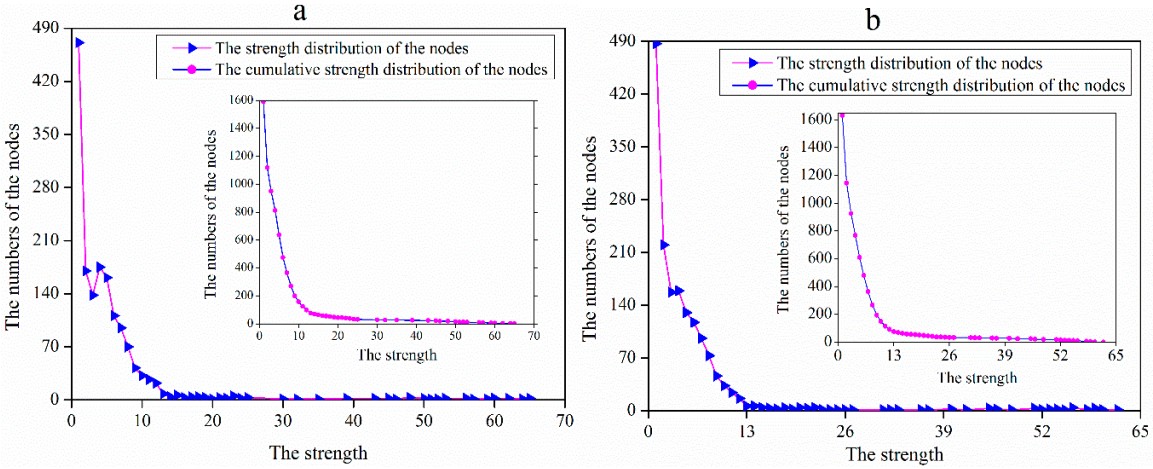

**Figure 6.** The number distribution of the strength and its cumulative number distribution of the CGSPFN in the whole period: (**a**) in NYH; (**b**) in GC.

According to Figure 6, most of the node strengths are small and only the minorities are large in the whole period, which shows a feature of the typical scale-free network. In addition, the important nodes mean that the node strengths are greater than or equal to 45 and they are different nodes. Meanwhile, some statistical indicators of the important nodes of the CGSPFN are given in Table 6.

**Table 6.** The statistical indicators of important nodes of the CGSPFN in NYH and GC.

| Statistical Indicators      Different Harbors | NYH | GC |
|---|---|---|
| The Number of the Nodes | 1592 | 1634 |
| The Number of the Important Nodes (The Strength ≥ 45) | 23 | 25 |
| The Percentage of the Important Nodes | 1.4447% | 1.5300% |
| The Sum of the Strength of the Important Nodes | 1255 | 1330 |
| The Percentage of the Sum of the Strength for the Important Nodes | 15.2010% | 16.1545% |
| The Average Contribution Rate of the Connections for the Important Nodes | 72.59% | 72.85% |

Based on the above analysis, it can be seen that the gasoline price fluctuation states in NYH and GC convert frequently and are complex, but the core fluctuation states are in the top 1.6% of the nodes,

which can be used to reflect the state and conversion relationship between the fluctuation states, and attempt approximate description of the essential characteristics of the gasoline price fluctuations in NYH and GC.

To some extent, the importance and influence of the nodes can be reflected by the node strengths in the whole network. Therefore, the important nodes are explored and we obtain the name, the strength and the temporal distribution feature about the nodes, as shown in Figure 7. In Figure 7, the symbol indicates the name of the node, the number on the line represents the weights between the two nodes, the red line donates the positive direction and the blue line donates the positive direction.

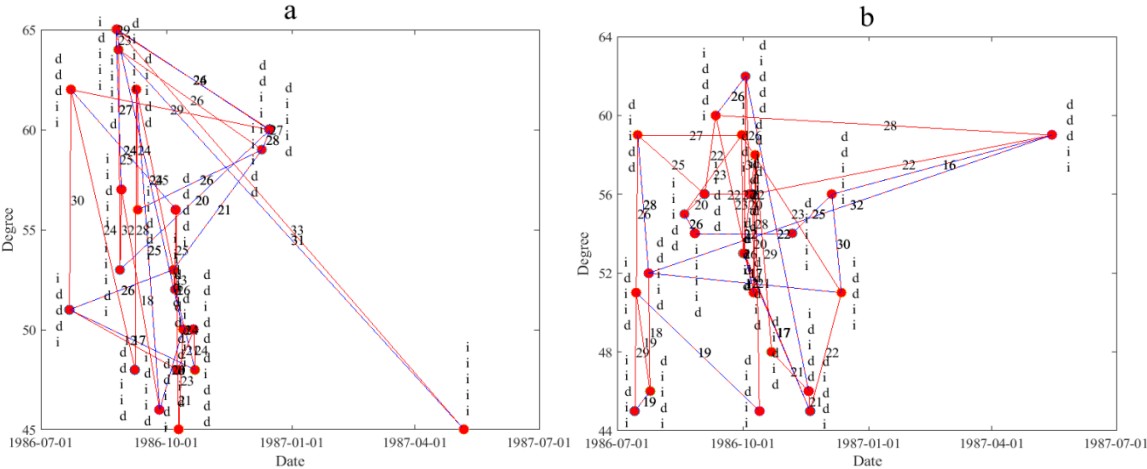

**Figure 7.** The temporal distribution feature of the important nodes of the CGSPFN in the whole period: (**a**) in NYH; (**b**) in GC.

In the terms of the temporal distribution feature of the important nodes in Figure 7, we can obtain the name, the strength, and the temporal distribution feature of the important nodes. As for the CGSPFN in NYH, the 23 important nodes are located in the first 112 nodes, the largest node strengths first appeared on 25 and 26 August, 1986, and they are located in the 30th and 31st node. Meanwhile, the value of the strength is 65, and the names of the nodes are 'idiii' and 'diiii', respectively. However, there are only 8 nodes and their strengths are 1 in the first 112 nodes; the earliest node is 'diIie', it is located in the 18th node and appeared on 7 August, 1986. As for the CGSPFN in GC, the 25 important nodes are located in the first 107 nodes; the largest node strengths first appeared on 3 October, 1986, and it is located in the 53rd node. Meanwhile, the value of the strength is 62, and the name of the node is 'diddd'. However, there are only 9 nodes and their strengths are 1 in the first 107 nodes; the earliest node is 'deide', it appeared on 23 February, 1987 and is located in 83rd node. The results show that the nodes with a large strength must be the nodes appearing in the earlier time, but the nodes appearing in the earlier time are not necessarily nodes with a large strength. Therefore, it can be seen that the links between the important nodes with a large strength are very close based on the weights of the connected edges among the nodes of the CGSPFN in NYH and GC.

In the terms of the average contribution rate of the connections among the important nodes of the CGSPFN in NYH and GC, it can be found that different CGSPFNs have obvious positive correlation features, i.e., the larger node strengths tend to be connected with a larger node strength. Although the conversion among the state of the CGSP is very frequent and the progress is complicated, the first 2.6% nodes (including the same nodes) can reflect their key fluctuations, which mean that the states of the CGSP fluctuations in the future may have already appeared in the early stage. Therefore, the fluctuation states and transformation relationships about the first 2.6% nodes are researched, and then the essential features of the CGSP can be found.

As is known to all, the network is a scale-free network if the strength distribution $p(s)$ can be fitted by the power-law distribution described by

$$p(s) = Cs^{-\gamma} \xrightarrow{\text{It takes logarithm on both sides at the same time.}} \ln p(s) = \ln C - \gamma \cdot \ln s \qquad (8)$$

where the $p(s)$ means the strength distribution, $C$ refers to the proportionality constant, $s$ is the strength, and $\gamma$ is the power law index. Meanwhile, the larger the power law index is, the stronger the power law distribution of the network is.

According to Formula (8), the strength distribution of the node is fitted by using the least squares method in different periods, then coefficient of determination ($R^2$), power law index ($\gamma$), and P-Value of the double logarithmic can be obtained, as shown in Table 7.

**Table 7.** The statistical indicators of the double logarithmic line about the CGSPFN.

| Different Harbors | Periods | $\gamma$ | $R^2$ | P-Value |
|---|---|---|---|---|
| NYH | The whole period | 1.6605 | 0.8490 | 0.0000 |
| | The SFP | 2.4588 | 0.9029 | 0.0000 |
| | The FFP | 2.1822 | 0.7216 | 0.0005 |
| GC | The whole period | 1.7216 | 0.8116 | 0.0000 |
| | The SFP | 2.4735 | 0.9297 | 0.0000 |
| | The FFP | 2.4985 | 0.8077 | 0.0000 |

According to Table 6, the fitting results have a high credibility and the CGSPFNs obey the power law distribution on the whole, which indicates that they are the scale-free network. In the terms of the scale-free network with higher levels of the power law distribution, the corresponding power law indexes are larger. Therefore, the level of the power law distribution about the CGSPFN in GC is higher than that in NYH in the whole period, which also illustrates the higher complexity of the CGSPFN in GC from another angle. As for the SFP and FFP, the power law indexes of the corresponding period in GC are larger than that in NYH, which illustrate that the strength of the power law distribution about the CGSPFN in GC is higher in the corresponding period and the power law distribution also shows a part of complexity and regularity. These results fully indicate that there are some differences in the two CGSPFNs.

### 3.4. Analysis of the Fluctuation Mode about the CGSP in NYH and GC

According to the relevant definition of the distance and average path length [42,43], the distance between any of two nodes in the two CGSPFNs is calculated by the Floyd algorithm [44], and the proportion of different lengths about the path is shown in Figure 8.

In Figure 8, we counted the distance between any of two nodes and the proportion of the same distance, then we obtained the proportion of the length of the path. The distance between any of two nodes means the time required from one mode to be converted to another, and the average path length indicates the conversion cycle among the fluctuation modes. Therefore, the conversion cycle of the fluctuation mode of the CGSPFN can be obtained by calculating the distance and the average path length among the nodes. According to Figure 8, the relevant statistical indicators of the fluctuation modes of the CGSPNs are given in the whole period, as shown in Table 8.

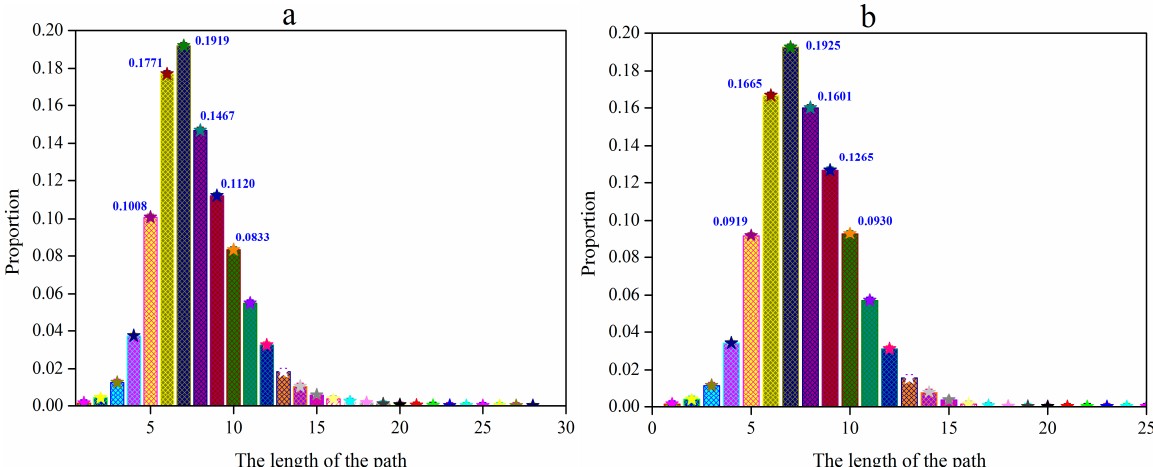

**Figure 8.** (**a**) the proportion of the length of different paths between any of two nodes in NYH; (**b**) the proportion of the length of different paths between any of two nodes in GC.

**Table 8.** The relevant statistical indicators of the fluctuation modes of the CGSPFNs in the whole period.

| Different Harbors | NYH | GC |
|---|---|---|
| The Diameter | 28 | 22 |
| The Average Path Length | 7.7483 | 7.7499 |
| The Proportion of the Distance between the Nodes (5–10) | 81.19% | 83.05% |
| The Conversion Cycle (Days) | 7–8 | 7–8 |

The relevant statistical indicators show that most of the fluctuation modes display short-range correlation and their conversions are more frequent with an average of 7–8. Comparing the diameter of the CGSPFN in NYH and GC, the diameter of the CGSPFN in GC is smaller than that in NYH, but they are basically the same conversion cycle. These results provide a basis for predicting the regularity of the periodic conversion of the CGSP.

In the SFP and FFP, we give the distribution of the length path among the fluctuation modes of the CGSPFNs, as shown in Figure 9.

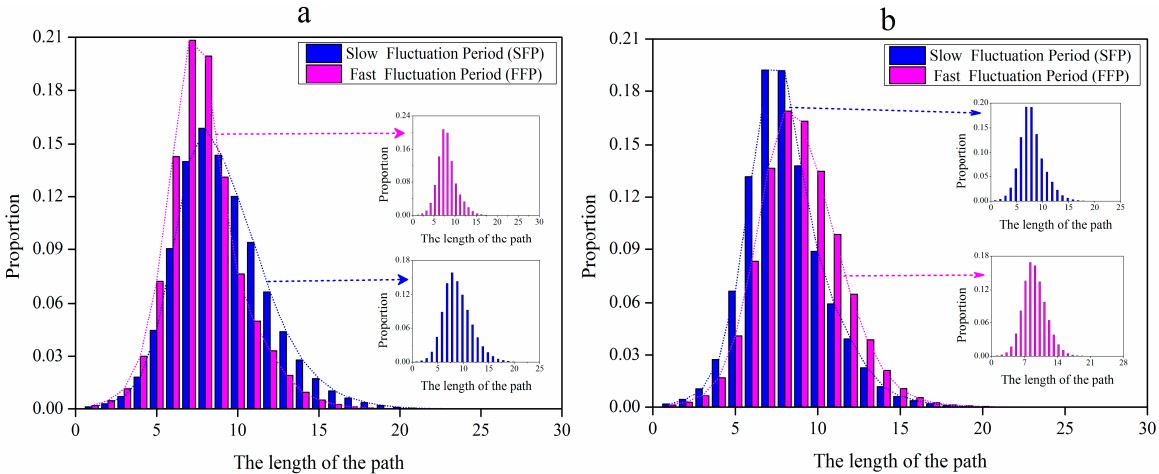

**Figure 9.** The proportion of the length of the path among the nodes of the CGSPFN: (**a**) in NYH; (**b**) in GC.

In order to display more clearly and intuitively, we give the relevant statistical indicators of the fluctuation modes of the CGSPFN in the SFP and FFP, as shown in Table 9.

**Table 9.** The relevant statistical indicators of the fluctuation modes of the CGSPN in the SFP and FFP.

| Harbors | Statistical Indicators | The SFP | The FFP |
| --- | --- | --- | --- |
| NYH | The Diameter | 29 | 23 |
| | The Average Path Length | 9.0660 | 7.8356 |
| | The Proportion of the Distance between the Nodes (6–11) | 74.69% | 80.77% |
| | Conversion Cycle (Days) | 9–10 | 7–8 |
| GC | The Diameter | 28 | 25 |
| | The Average Path Length | 8.9286 | 8.0585 |
| | The Proportion of the Distance between the Nodes (6–11) | 78.52% | 80.19% |
| | Conversion Cycle (Days) | 8–9 | 8–9 |

According to Table 9, the diameter of the CGSPFNs in NYH in the SFP is larger than that in GC, but it is the opposite in the FFP, which means that the path length had changed, i.e., the change of the conversion cycle. In the SFP, the conversion cycle of the fluctuation modes in GC is shorter than that in NYH, which means that the conversion is more frequent. However, it is opposite in the FFP. Moreover, there is a little difference in the value of the statistical indicators, which illustrates that the fluctuation modes of the two CGSPFNs have an approximate conversion cycle in the corresponding period. Therefore, we can predict the time of the conversion among the nodes by identifying the conversion cycle, which is beneficial to decrease the risk of the CGSP fluctuation for the decision makers.

*3.5. Analysis of the Betweenness about the CGSPFNs in NYH and GC*

According to the definition and calculation formula of the betweenness of the node about the network [32,34,45], we explore the fluctuation mode about the CGSP in the whole, and the evolutionary relationships between the betweenness and the strengths about the CGSPFN in the whole period are analyzed, as shown in Figure 10.

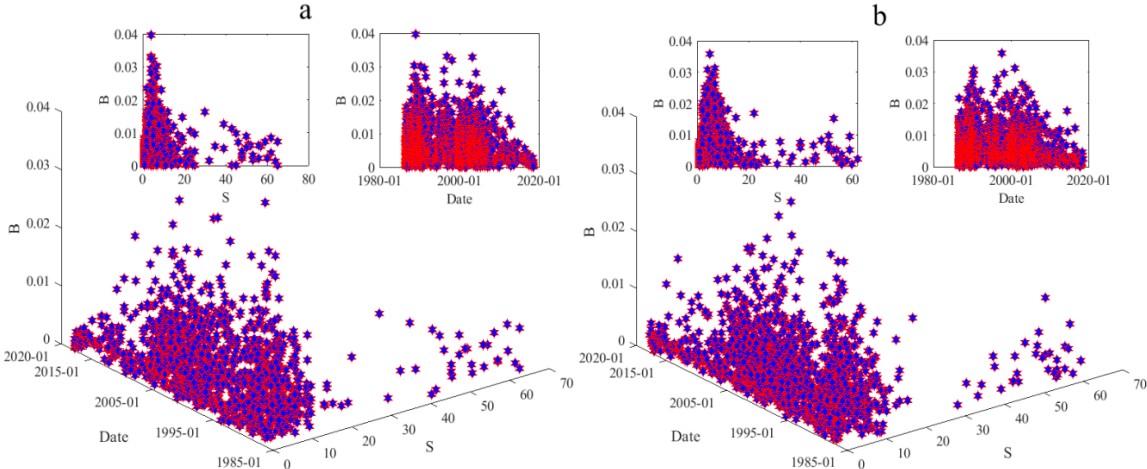

**Figure 10.** The evolutionary relationships between the betweenness and node strengths of the CGSPFNs with the time in the whole period: (**a**) in NYH; (**b**) in GC. Note: The B refers to the betweenness, and the S is the strength in Figures 10, 11 and 14.

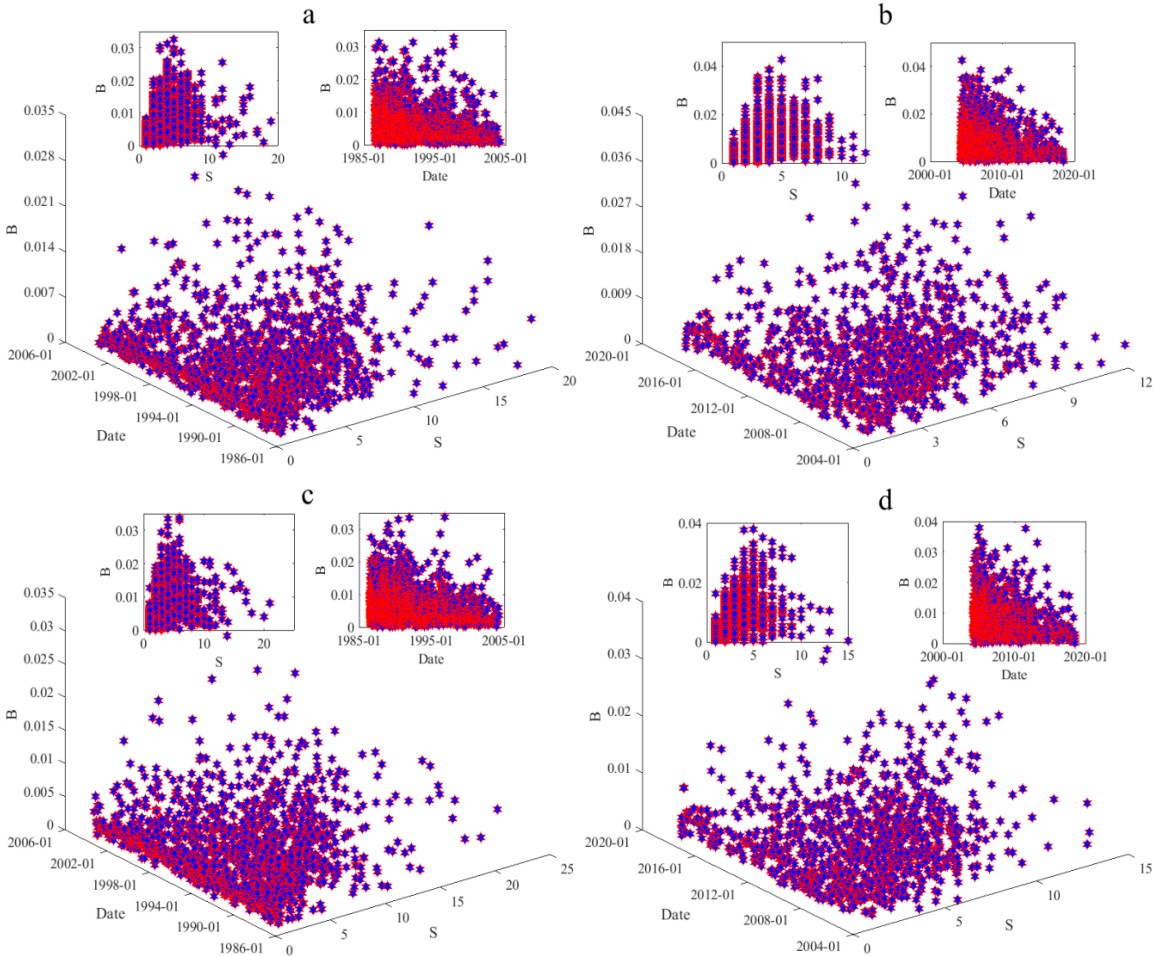

**Figure 11.** The evolutionary relationships between the betweenness and node strengths of the CGSPFN with the time: (**a**) in the SFP in NYH; (**b**) in the FFP in NYH; (**c**) in the SFP in GC; (**d**) in the FFP in GC.

According to Figure 10, there are less nodes with a larger betweenness and the larger betweenness of the nodes have less strength, and it appeared in an earlier time. Meanwhile, the larger node strengths have less betweenness and it appeared in an earlier time. In addition, the largest betweenness and strength in NYH are larger than that in GC, which illustrates that the nodes of the CGSPFN in NYH have stronger ability of the connectivity than that in GC.

As shown in Figure 10, we count the betweenness of the nodes of the CGSPFN in descending order of its value in NYH and GC, then we list the first six and the corresponding strengths, the name and the first time that the node appears, as shown in Table 10.

In terms of the relationship between the betweenness and the strength in the whole period, it can be seen that the betweenness of the nodes is larger but the strength is smaller in Figure 8 and Table 10, which illustrates that the nodes with less strength have important connectivity and influence in the CGSPFN.

In order to analyze the relationships between the betweenness and the strength under the temporal distribution in the SFP and FFP, their evolutionary relationships are researched, as shown in Figure 11.

**Table 10.** The first six betweenness of the nodes of the CGSPFN in descending order of its value.

| NYH | | | |
|---|---|---|---|
| **The Betweenness** | **The Strength** | **The Name** | **The First Appeared Time** |
| 0.039756 | 4 | 'didDi' | 7 December, 1988 |
| 0.033083 | 4 | 'IDidi' | 26 September, 1996 |
| 0.032756 | 4 | 'IIDid' | 20 February, 2003 |
| 0.032041 | 5 | 'IiiID' | 23 January, 1998 |
| 0.030411 | 6 | 'iiDdD' | 19 July, 1988 |
| 0.030102 | 7 | 'dIdii' | 17 April, 1990 |
| **GC** | | | |
| **The Betweenness** | **The Strength** | **The Name** | **The First Time Appeared** |
| 0.035844 | 5 | 'Iiddi' | 28 July, 1997 |
| 0.031236 | 7 | 'IIDId' | 4 January, 2001 |
| 0.030703 | 4 | 'dIidd' | 19 April, 1990 |
| 0.029426 | 6 | 'ddDDd' | 3 March, 1998 |
| 0.028912 | 6 | 'iDddd' | 31 May, 1990 |
| 0.028882 | 7 | 'iiIdI' | 12 March, 1990 |

As for the CGSPFN in NYH in Figure 11a,b, the first six betweenness of the nodes in the SFP are 0.03258, 0.0313, 0.03119, 0.03038, 0.02991 and 0.02943, respectively, and the corresponding strengths are 5, 4, 4, 3, 5 and 3. In the FFP, the first six betweenness of the nodes are 0.03367, 0.03341, 0.03309, 0.03133, 0.02939 and 0.02865, respectively, and the corresponding strengths are 5, 4, 4, 3, 5 and 4. As for the CGSPFN in GC in Figure 11c,d, the first six betweenness of the nodes in the SFP are 0.03258, 0.0313, 0.03119, 0.03038, 0.02991 and 0.02943, respectively, and the corresponding strengths are 5, 4, 4, 3, 5 and 3. In the FFP, the first six betweenness of the nodes are 0.03367, 0.03341, 0.03309, 0.03133, 0.02939 and 0.02865, respectively, and the corresponding strengths are 5, 4, 6, 7, 5 and 5. By the comparative analysis, the betweenness of the nodes in the FFP is larger than that in SFP, which illustrates that the impact of some unexpected emergencies enhances the intermediary betweenness of the nodes and indicates a higher influence. Meanwhile, the betweenness of the nodes shows a downward trend in the SFP, i.e., the impact of the nodes decreases. Comparing the relationships between the betweenness and the strength, the nodes with the smaller strength play a major role of the connectivity in the CGSPFN, but the nodes connect with other nodes by the nodes with the larger strength. When the betweenness is higher and there is a tendency with the continual increase, which illustrates that other nodes must pass through this node so as to connect with the other nodes, then this period is an important period. According to the results, we can give an effective prejudgment of the CGSP in the next period.

*3.6. Analysis of the Clustering Features between the Fluctuation Modes*

According to the definition and calculation formula of the clustering coefficient [34,46] and average clustering coefficient [47] of the nodes about the network, the evolutionary relationships between the clustering coefficient and the strength, the first appeared time of the nodes of the CGSPFN, are analyzed in NYH and GC, as shown in Figure 12.

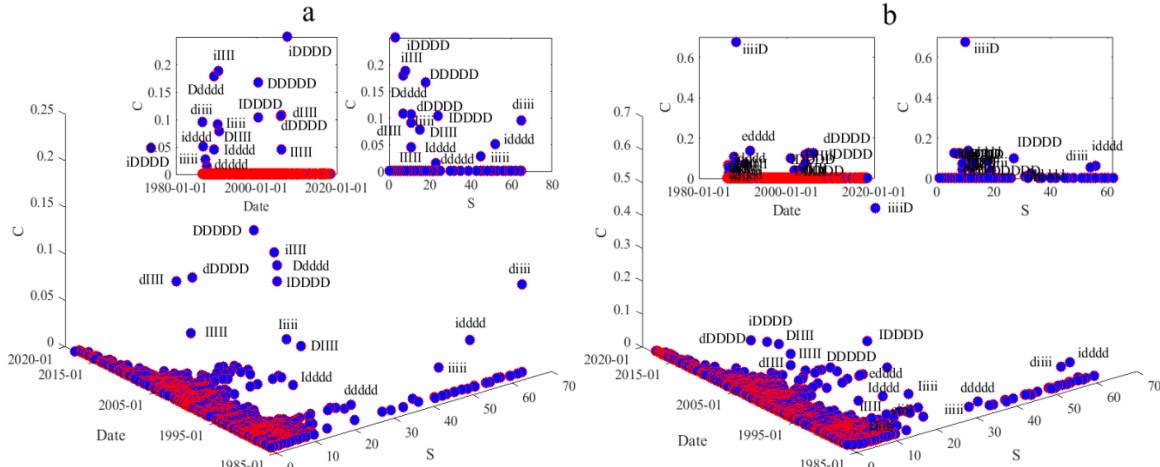

**Figure 12.** The evolutionary relationships between the clustering coefficient and the strength, the first appeared time of the nodes of the CGSPFN in the whole period: (**a**) in NYH; (**b**) in GC. Note: The C refers to the clustering coefficients and the S is the strength in Figures 12–14.

According to Figure 12a, there are only 15 nodes of the CGSPFN whose clustering coefficients are not 0 in NYH. According to the descending order of the clustering coefficient s of the nodes, they are 0.25, 0.1875, 0.17857, 0.16667, 0.10714, 0.10606, 0.10417, 0.09488, 0.09091, 0.07778, 0.05049, 0.04545, 0.04545, 0.02778 and 0.01450, respectively. The corresponding names are 'iDDDD', 'iIIII', 'Ddddd', 'DDDDD', 'dIIII', 'dDDDD', 'IDDDD', 'diiii', 'Iiiii', 'DIIII', 'idddd', 'Idddd', 'IIIII', 'iiiii' and 'ddddd'. Meanwhile, the corresponding strengths are 3, 8, 7, 18, 7, 11, 24, 65, 11, 15, 52, 11, 11, 45 and 23. As shown in Figure 12b, there are 18 nodes of the CGSPFN whose clustering coefficients are not 0 in GC. According to the descending order of the clustering coefficients of the nodes, they are 0.675, 0.13636, 0.125, 0.125, 0.11667, 0.10606, 0.09876, 0.08333, 0.07692, 0.07407, 0.06667, 0.06026, 0.05324, 0.05208, 0.04167, 0.039474, 0.02273 and 0.01075, respectively. The corresponding names are 'iiiiD', 'edddd', 'iDDDD', 'dDDDD', 'DIIII', 'Idddd', 'IDDDD', 'Iiiii', 'IIIII', 'iIIII', 'eiiii', 'idddd', 'diiii', 'dIIII', 'Diiii', 'DDDDD', 'ddddd' and 'iiiii'. Meanwhile, the corresponding strengths are 10, 11, 8, 6, 10, 11, 27, 18, 13, 9, 10, 56, 54, 12, 9, 19, 33 and 31. On the basis of the above results, the clustering coefficients and their average are all smaller in NYH and GC, where the average clustering coefficients are 0.00097, 0.00120, respectively; i.e., the probability that two nodes connected to the same nodes are also connected to each other is small. However, the number of the clustering coefficients that are not 0 is more than that in NYH, and the average clustering coefficients of the CGSPFN in GC are larger than that in NYH, which all illustrate that the CGSPFN in GC is more closely tied than that in NYH. As for the nodes where the clustering coefficients are not 0, most of the strength of which are smaller, but there are a few nodes with a large strength, which show that the CGSPFN is not completely random in NYH and GC, and has the features of the community. The conclusion can be found in Figure 4, where the nodes with the same color present a community. According to these results, we can find that the obvious clustering features in the CGSPFN may occur not only in small communities, but also in large ones. In addition, we explore the temporal distribution characteristics of the nodes where the clustering coefficients are not 0, as shown in Table 11.

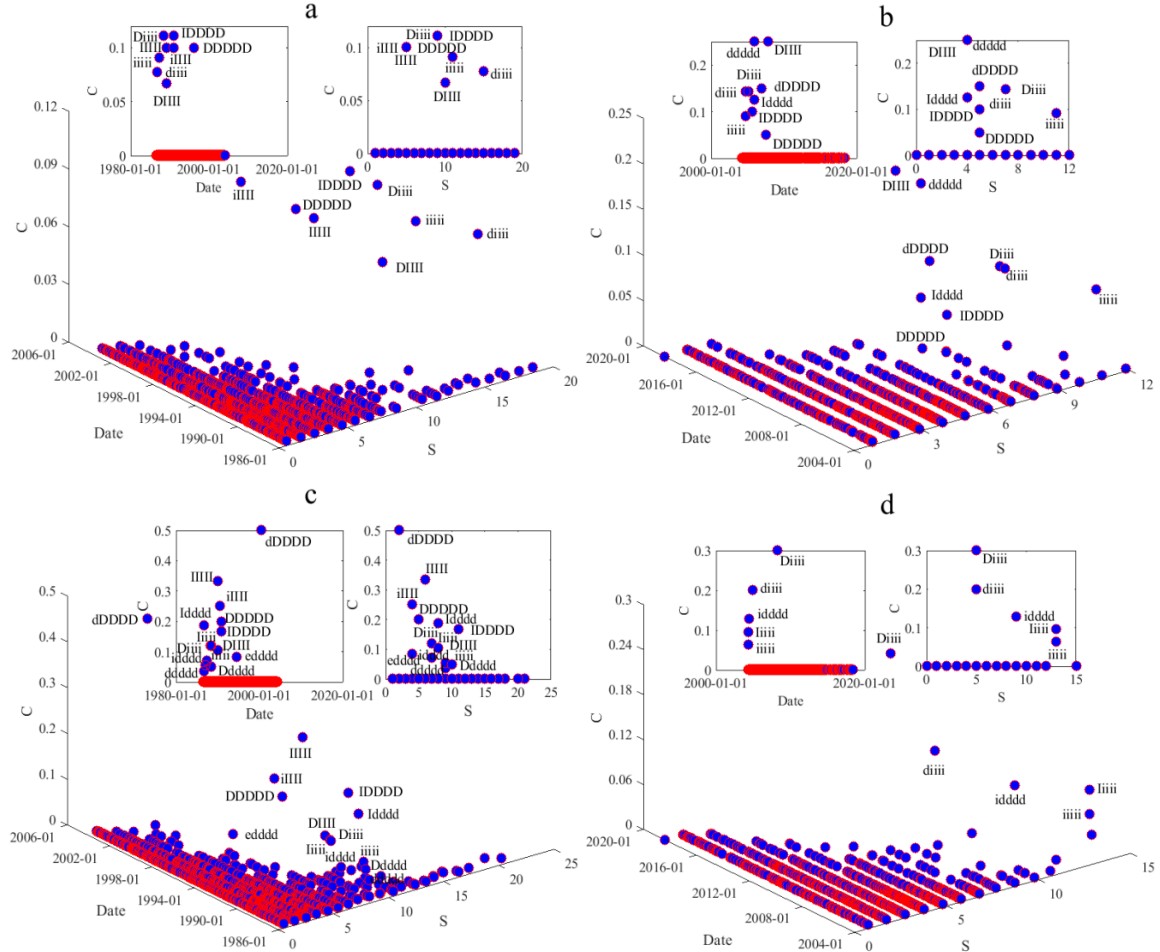

**Figure 13.** The evolutionary relationships between the clustering coefficient and the strength, the first appeared time of the nodes of the CGSPFN: (**a**) in the SFP in NYH; (**b**) in the FFP in NYH; (**c**) in the SFP in GC; (**d**) in the FFP in GC.

**Table 11.** The temporal distribution characteristics of the clustering coefficients about the nodes of the CGSPFN in the whole period in NYH and GC.

| Different Harbors | NYH | GC |
|---|---|---|
| The First Appeared Time | 26 August, 1986 | 26 August, 1986 |
| The Last Appeared Time | 19 September, 2007 | 15 March, 2006 |

According to Table 11, we can find that the first appeared time in NYH is the same as in GC, which illustrates that the two CGSPFNs have similar time distribution characteristics in a small-time scale; but their differences increase when the time scale is expanded, which shows that the clustering of the CGSP fluctuations may be reflected not only on a large time scale, but also on a small-time scale.

In the SFP and FFP, we give the evolutionary image of the clustering coefficients of the nodes, and the clustering coefficients are not 0, as shown in Figure 13.

As for the CGSPFN in NYH, there are only 8 nodes whose clustering coefficients are not 0 in the SFP. According to the descending order of the clustering coefficients of the nodes in the SFP, they are 0.11111, 0.11111, 0.1, 0.1, 0.1, 0.09091, 0.07778 and 0.06667, respectively. The corresponding names are 'Diiii', 'IDDDD', 'IIIII', 'DDDDD', 'iIIII', 'iiiii', 'diiii' and 'DIIII', then the corresponding strengths are 9, 9, 5, 5, 5, 11, 15 and 10. Meanwhile, there are only 9 nodes whose clustering coefficients are not 0 in the FFP. According to the descending order of the clustering coefficients of the nodes in the FFP,

they are 0.25, 0.25, 0.15, 0.142857, 0.14286, 0.125, 0.1, 0.09091 and 0.05, respectively. The corresponding names are 'ddddd', 'DIIII', 'dDDDD', 'diiii', 'Diiii', 'Idddd', 'IDDDD', 'iiiii' and 'DDDDD', then the corresponding strengths are 4, 4, 5, 7, 7, 4, 5, 11 and 5. As for the CGSPFN in GC, there are 15 nodes whose clustering coefficients are not 0 in the SFP. According to the descending order of the CCs of the nodes in the SFP, they are 0.5, 0.33333, 0.25, 0.2, 0.1875, 0.16667, 0.11905, 0.11905, 0.10417, 0.08333, 0.07143, 0.05556, 0.05 and 0.03704, respectively. The corresponding names are 'dDDDD', 'IIIII', 'iIIII', 'DDDDD', 'Idddd', 'IDDDD', 'Iiiii', 'Diiii', 'DIIII', 'edddd', 'idddd', 'Ddddd', 'iiiii' and 'ddddd', then the corresponding strengths are 9, 9, 5, 5, 5, 11, 15 and 10. Meanwhile, there are 5 nodes whose clustering coefficients are not 0 in the FFP. According to the descending order of the clustering coefficients of the nodes in the FFP, they are 0.3, 0.2, 0.12963, 0.09615 and 0.06410, respectively. The corresponding names are 'Diiii', 'diiii', 'idddd', 'Iiiii' and 'iiiii', then the corresponding strengths are 5, 5, 9, 13 and 13, with average clustering coefficients shown in Table 12.

**Table 12.** The average clustering coefficients of the nodes of the CGSPFN in the SFP and FFP.

| Periods | Harbors | The Number of the Clustering Coefficients | The Average Clustering Coefficients |
|---------|---------|-------------------------------------------|-------------------------------------|
| The SFP | NYH | 8 | 0.00046 |
|         | GC  | 15 | 0.00138 |
| TheFFP  | NYH | 9 | 0.00112 |
|         | GC  | 5 | 0.00065 |

By analyzing the clustering coefficients in the SFP and FFP, we can find that the number of the nodes whose clustering coefficients are not 0 in GC is more than that in NYH in the SFP, and it is opposite in the FFP, which illustrate that the CGSPFN structure of the GC is closer in the SFP, but the CGSPFN structure of the NYH is closer than that in GC in the FFP. These results all show that the CGSP of different harbors have more complex characteristics in different periods.

## 4. The Comprehensive Discussion

Combined with the above analysis of the temporal distribution characteristics about the strength, betweenness and clustering coefficient of the nodes in NYH and GC, the evolutionary relationships among them are explored in the whole period. Therefore, we choose the core nodes with a larger strength, betweenness and clustering coefficient, whose evolutionary relationships among them are shown in Figure 14a,c, and the temporal distribution characteristics, of which are obtained when they first appear, are shown in Figure 14b,d.

In Figure 14, S, C and B present the strength, clustering coefficient and betweenness of the core nodes, respectively. Meanwhile, the larger strength, clustering coefficient, and betweenness are displayed by the blue '○', the green '◇', the red '△', respectively. According to Figure 14a,c, the CGSPFN all shows that the core nodes with the larger strength have smaller betweenness and clustering coefficients, the core nodes with the larger betweenness have smaller strength and clustering coefficients, and the core nodes with the larger clustering coefficients have smaller betweenness and strength. Based on the above analysis, it can be seen that the node of the CGSPFN in NYH and GC has a smaller clustering coefficient, average path length, and betweenness, which is different from the random network and chaotic network.

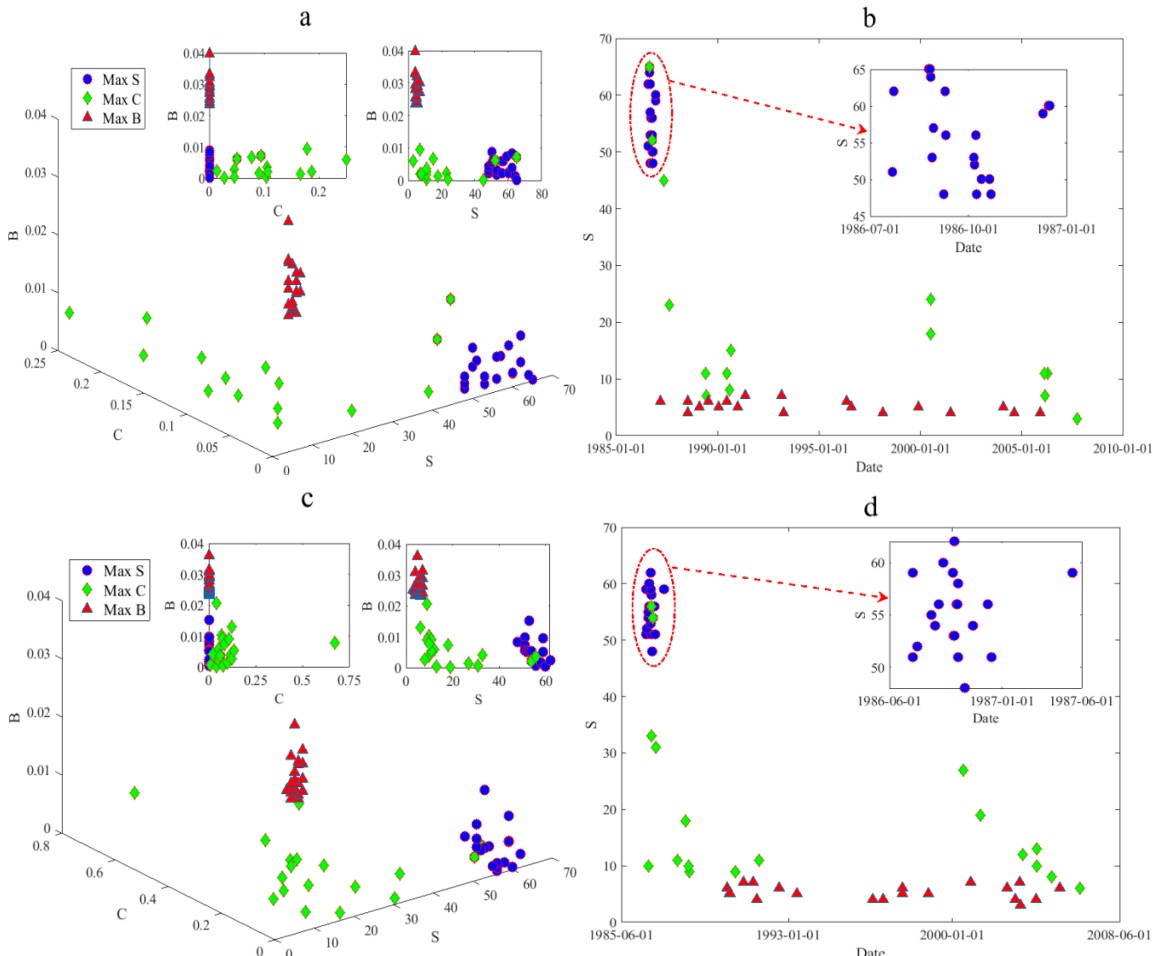

**Figure 14.** The temporal distribution characteristics of the larger strength, betweenness and clustering coefficient of the core nodes: (**a**,**b**) in NYH; (**c**,**d**) in GC.

In terms of the first appeared time with them in Figure 14b,d, their concentrated periods are shown in Table 13.

**Table 13.** The concentrated period of the core nodes of the CGSPFN in NYH and GC.

| Different Harbors | The Larger Indicators | The Concentrative Period |
|---|---|---|
| NYH | The Strength | July 1986 to January 1987 |
| | The Clustering Coefficient | July 1986 to October 2007 |
| | The Betweenness | July 1986 to January 2006 |
| GC | The Strength | July 1986 to June 1987 |
| | The Clustering Coefficient | July 1986 to May 2006 |
| | The Betweenness | January 1990 to May 2005 |

According to the time distribution characteristics when the core node first appears in the CGSPFN of the NYH and GC, Figure 14b,d and Table 13 show that the core nodes with a larger strength (see the blue '○') first appear at an earlier time, which means that the core nodes with a larger strength in the CGSPFN of the NYH and GC are the nodes which appeared at an earlier time. The core nodes with a larger clustering coefficient (see the green '◇') first appear in a relatively dispersive way, but still incline to the early period. The core nodes with a larger betweenness (see the red '△') first appear in the most decentralized time before 2006. These results show that the gasoline prices are in a transitional

period when the larger indicators appear and have a rising trend; identifying the transitional period will help the decision maker to grasp the regularity of the changes of the gasoline prices.

## 5. Summaries and Implications

In order to reveal the fluctuation mechanism of the gasoline prices, this paper defined the fluctuation modes by the coarse-grained method based on the CGSP series in NYH and GC. We have converted the fluctuation series into the characters by means of the sliding window, where five symbol series were used as a fluctuation mode, and one day was used as a step to slide in the data window. The periods of the time series data are divided into the SFP and FFP based on the different fluctuation states in NYH and GC. As for the different periods, the mode was defined as the node of the network, the direction and times of the transformation among the mode was defined as the edge and weight, respectively. Then, the CGSPFN of the NYH and GC were constructed in the different periods. In the terms of the CGSPFN in NYH and GC, the evolutionary rules of the new nodes in the CGSPFNs were analyzed, and the strength and distribution, average shortest paths, conversion cycle, betweenness, clustering coefficient of the nodes were calculated in different periods. In addition, we identified the important nodes and the time appeared and calculated the similarity of the node strength in different periods between the NYH and GC. The results are obtained as following:

(1) The similarity of the CGSPFN between the NYH and GC is 0.7405 in the whole period. However, the similarity of the CGSPFN between the NYH and GC is 0.4986 in the SFP, but it is 0.3069 in the FFP. These present that there is a higher similarity and the interdependence in the whole period in the two harbors, but there is a complex dependence relationship in FFP.

(2) Although there is a complicated nonlinear character for the gasoline price fluctuation, the cumulative times are not nonlinear and show a high linear growth trend when the new node appears in the CGSPFN, which is different in different periods. As for the CGSPFN in NYH, the cumulative time when the new nodes appear in the SFP is longer than the SSF, but for the CGSPFN in GC, it is contrary.

(3) The strengths and their distributions of the nodes show that the core fluctuation state of the CGSPFNs is reflected in the first 2.6% nodes, which displays the importance of some nodes in the CGSPFNs. For the important nodes of the CGSPFN, the average contribution rate of the connections is 72.59% in NYH, but the average contribution rate of the connections is 72.85%. Therefore, there is an obvious positive correlation feature, i.e., the larger node strengths tend to be connected with a larger node strength. From the temporal distribution feature of the important nodes of the CGSPFN in the whole period, the results that the nodes with a large strength must be the nodes appearing in the earlier time, but the nodes appearing in the earlier time are not necessarily nodes with a large strength. In addition, the CGSPFNs are the scale-free network, and the CGSPFN in NYH is more heterogeneous than in GC. Meanwhile, the value of the power law index is different in different periods. As for the SFP and FFP, the power law indexes of the corresponding period in GC are larger than that in NYH.

(4) As for the whole period, the diameter of the CGSPFN in NYH is more than that in GC, and the conversion cycle among the fluctuation modes is the same, which is 7–8 days, which presents a short-range correlation. As for the CGSPFN in NYH and GC, the diameter in the SFP is more than that in FFP, the conversion cycle in NYH is more than that in GC in the SFP, but which is in NYH less than that in GC in the FFP. According to these, we can predict the time of the conversion among the nodes by identifying the conversion cycle, which is beneficial to decrease the risk of the CGSP fluctuation for the decision makers.

(5) The evolutionary relationships between the betweenness and node strengths of the CGSPFNs with the time in the whole period reveal that the strength of nodes with large betweenness are small, whether it is in NYH or in GC. Meanwhile, the nodes with a smaller strength act as the main intermediary function in the CGSPFN. When the node with a small strength appears, it means that the period is in a transition period. Therefore, we can effectively predict the fluctuation state of the CGSP in the next period and an adjustment strategy will be taken by the decision-makers when the node is identified and analyzed.

(6) The average clustering coefficients of the CGSPFN in NYH and GC are 0.00097 and 0.00120, respectively, and the former is closer than the latter, which show that the CGSPFN is not completely random in NYH and GC, and has the features of the community. According to the evolutionary relationships between the clustering coefficient and the strength with the first appeared time of the nodes of the CGSPFN in the whole period, some references and help can be provided for studying the community of the gasoline prices in the future.

Gasoline is a depletable resource, in which demand-supply is rarely in a competitive equilibrium market [8,48,49]. Meanwhile, the gasoline market that is affected by many uncertain factors is a relatively complex system, such as the cost of crude oil, the cost of producing products, the marine climate, the cost of distribution and transportation of products, and the spot and future price fluctuations of the gasoline, which lead to frequent fluctuations of the CGSP. Based on the research of this paper, we provide some theoretical references for the prejudgment of the CGSP market, which have a certain guiding significance to research the systematic risk behavior [50] and guide the economic life, such as the asymmetry of gasoline prices to oil price shocks and policy uncertainty [51], the effects of oil prices to gasoline prices [52,53], and the gasoline prices and fuel efficiency [54,55]. Furthermore, the core relationships for identifying the complex relationships among the multiple elements will be focused on by means of constructing the interrelationships of the multiple elements in future research.

**Author Contributions:** G.Z. designed and completed the experiments, and then wrote the manuscript. L.T. guided and enriched the whole research. W.Z. and X.Y. participated in the discussion and provided some advices. X.Y., B.W. and Z.Z. completed the English revision of the manuscript. All authors have read and agreed to the published version of the manuscript.

**Funding:** This research is supported by grants from the National Natural Science Foundation of China (No: 71690242, 91546118, 11731014, 51976085), the Social Science Foundation of Jiangsu Province (No: 18EYB020), Qing Lan Project of Jiangsu Province (SJ201812), Jiangsu Key Lab for NSLSCS (201904), Soft science project of Jiangsu (BR2019027), and Postgraduate Research & Practice Innovation Program of Jiangsu Province (KYCX18_2256).

**Conflicts of Interest:** The authors declare no conflict of interest.

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
