# Peer review of "A Study on the Similarities and Differences of the Conventional Gasoline Spot Price Fluctuation Network between Different Harbors"

_sustainability, doi:10.3390/su12020710_

Round 1

Reviewer 1 Report

The article is written thoroughly and is very interesting.
The authors described the empirical research in detail in the article. They also described the results of the study in detail. All conclusions and analyzes are supported by graphs, tables and summaries.

Reviewer 2 Report

Author(s) analysed the Similarities and Differences of the Conventional Gasoline Spot Price Fluctuation Network between Different Harbors. The topic itself is very important but in the current states, it did not reflect the issues in hand. In the current form, the methodology adopted is not enough explored. Using complex mathematics does not make per se good science, you always have to provide a very sound conceptual framework before measuring.  so all of these mathematical and statistical formulae have some pros and cons and results are vary with the method used. So we put the data in the black box and get some filtered data and no theory behind to explain these differences because some of them are pure statistical methods. As for the present paper, Author(s) used complex mathematical formulae that have not considered any economic, social, or environmental aspects when measuring the similarities and differences of the  Conventional Gasoline Spot Price Fluctuation Network between Different Harbors. One of the main drivers of gasoline prices is Crude oil prices continue to be the main driver of gasoline prices. There are so many other factors that influence the gasoline prices, such as demand and supply conditions differ across the nation and in particular, regional differences in ease of access to gasoline supplies from refineries or pipelines, as well as some boutique fuel requirements, appear to influence gasoline prices. For example, boutique fuel gross product margins in the Gulf are not significantly more variable than those for conventional gasoline. Gasoline prices vary over time and among states and regions. In addition to differences in the state, other factors contribute to regional differences in gasoline prices, including distance from the supply, supply disruptions, and retail competition and operating costs. Environmental programs add to the cost of production, storage, and distribution. Besides the above, the communication and presentation of results should also be improved. Specific suggestions are outlined below.

Some acronyms are missing, such as page 2, line 91, EIA, is missing, etc.  Furthermore, once defined in the first place, the acronyms should not be repeated.  What new insight or analysis does the article present? In what way does it contribute to the existing literature? It should be spelled out early in the introduction section. As the main finding suggests that  On page 2, line 79-81, authors discuss as “However, there are few studies on the similarities and differences of the fluctuation of the gasoline prices between different harbors, which are explored to make some valuable results by the complex network theory and random matrix theory” which studies, clearly elaborates here and find out the gap with a critical review of past literature. What are the new main innovations of this paper? Any gap in the existing literature which fillup this study,  a vast literature is available on this issue such as, Gasoline Price Changes: The Dynamic of Supply, Demand, and Competition. Federal Trade Commission, 2005 (A Report). Huan Chen, Lixin Tian, Minggang Wang and Zaili Zhen (2017). Analysis of the Dynamic Evolutionary Behavior of American Heating Oil Spot and Futures Price Fluctuation Networks. Sustainability 2017, 9, 574; doi:10.3390/su9040574 Du R, Dong G, Tian L, Wang M, Fang G, Shao S (2016) Spatiotemporal Dynamics and Fitness Analysis of Global Oil Market: Based on Complex Network. PLoS ONE 11(10): e0162362. https://doi.org/10.1371/journal.pone.0162362 An HZ, Gao XY, Fang W, Ding YH, Zhong WQ. Research on patterns in the fluctuation of the co-movement between crude oil futures and spot prices: A complex network approach. Appl. Energy. 2014; 136: 1067–1075. Wang, M.G.; Chen, Y.; Tian, L.X.; Jiang, S.M.; Tian, Z.H.; Du, R.J. Fluctuation behavior analysis of international crude oil and gasoline price based on a complex network perspective. Appl. Energy 2016, 175, 109–127 Wang, M.G.; Tian, L.X. From time series to complex networks: The phase space coarse graining. Phys. A Stat. Mech. Appl. 2016, 461, 456–468. On-Page 3, Table 5, regression estimates for GC harbor y = 19417x–179.0398 in the period of SFP the slope coefficient is much higher than the respective NYH harbor.  It should be 1.9417x or if it is correct then give more explanation about this large difference. It should also show the significance of the variables in these estimated regression equations, such as shows t-values, standard errors or p-values. Correlation matters but if one variable is insignificant then it means it has no relevance in determining the dependent variable. Page 15, Table 7, correlation coefficient ( g ) values of both harbors for all periods are greater than 1 which are statistically not correct, as the correlation coefficient values range between -1.0 and 1.0. If a calculated number greater than 1.0 or less than -1.0 means that there was an error in the correlation measurement. The correlation coefficient is a statistical measure that calculates the strength of the relationship between the relative movements of two variables. The conclusions section would benefit from the simplification and mainstreaming of the main findings. A wrap-up of the main achievements is also important. Some methods are stressed, whilst some others are not mentioned/used. For instance, besides statistical analysis, it is important to expose clearly the choices behind the methodology, the complex network theory and random matrix theory, etc. (if any). No single reference provided in the results and discussion section. Compare and contrast the study results with previous studies, which is the main weakness of the study, as it shows that there is no such study exist in the past. If this is the case, then it should briefly spell out early in the introduction section with a critical discussion on the choice of the methodology adopted.

Reviewer 3 Report

The authors study the structure of Gasoline Spot Price between different harbors. I agree with the authors that research on Gasoline market is still scarce and the topic is therefore highly relevant. However, the paper raises a couple of questions that should be clarified. In particular,

The current version of the abstract can be improved in order to be more informative. In line 36 it is not so clear to me what the authors want to say with “macroeconomic fluctuations”? Is this related to the state of the economy? Is this paper related to the Law of One Price? The presentation of the data can be improved. Are the data on a daily basis? What is EIA? etc… On line 101, the authors define the term ΔP. This is somehow a return series, why you do not use log-returns which more standard? Why note volatility series? Are new series stationary? I am puzzled with the Tables 5-7. What exactly is the correlation coefficient R^2? Is this related to the linear regression? Is this the standard R^2? On Table 7, it is not clear to me what exactly is the γ parameter? Where are the standard error of this particular parameter? The R^2 is very high, is this usual? Not so sure here if the results are purely random or not. I am puzzled with the text in lines 605 to 607. In particular with the term systemic risk behaviour? What exactly do you mean here? The results section (section 3) reads like a laundry list of different tests/steps. It is very difficult to wade through this material and absorb the bigger picture. On the whole, I think the authors should focus on the key comparisons and relegate much of the remaining material to footnotes and appendices. I doubt that someone who is not already familiar with Networks would be able to figure out what the authors are actually doing. They introduce numerous specialist terms without defining them, e.g. “EIA”. Again, a specialist in this topic will clearly know what this is, but not a general audience. Furthermore, the authors used too many acronyms. I must say that I get quite tired by reading this paper.

Round 2

Reviewer 2 Report

All comments are incorporated. 

Reviewer 3 Report

In general, I am happy with the revision. However, I wasn't convinced by the explanation provided by the authors in comment 4. Regarding my previous comment (see Point 4), please use the proper definition for the R^2 see e.g. https://www.investopedia.com/terms/r/r-squared.asp  
